# Investigating Adherence to COVID-19 Vaccination and Serum Antibody Concentration among Hospital Workers—The Experience of an Italian Private Hospital

**DOI:** 10.3390/vaccines9111332

**Published:** 2021-11-16

**Authors:** Guglielmo Forgeschi, Giuseppe Cavallo, Chiara Lorini, Fiamma Balboni, Francesca Sequi, Guglielmo Bonaccorsi

**Affiliations:** 1Istituto Fiorentino di Cura e Assistenza S.p.A. (IFCA), Via del Pergolino 4, 50139 Florence, Italy; g.forgeschi@giomi.com (G.F.); fiamma.balboni@hotmail.it (F.B.); f.sequi@giomi.com (F.S.); 2Department of Health Science, University of Florence, Viale GB Morgagni 48, 50134 Florence, Italy; chiara.lorini@unifi.it (C.L.); guglielmo.bonaccorsi@unifi.it (G.B.)

**Keywords:** SARS-CoV-2, COVID-19, vaccine hesitancy, vaccine adherence, anti-S antibodies

## Abstract

SARS-CoV-2 transmission has been high, especially among healthcare workers worldwide during the first wave. Vaccination is recognized as the most effective approach to combat the pandemic, but hesitation to get vaccinated represents an obstacle. Another important issue is the duration of protection after administration of the full vaccination cycle. Based on these premises, we conducted a study to evaluate vaccination adherence and the anti-S antibodies levels among hospital workers, from January to March, 2021. To assess adherence, an anonymous questionnaire was used. Anti-S antibody levels were obtained from the monitoring serological sample database. In total, 56.2% of the unvaccinated people did not report a previous infection from COVID-19. Among those who have not been vaccinated, 12.5% showed distrust against the vaccine, 8.3% stated to have received contraindications to the vaccination, and 6.3% did not report any choice. Analyzing anti-S antibody levels, only one person was found to have a value below the lower cut-off, two weeks, and three months after receiving their second dose. One was below the cut-off after two weeks, and then above the same cut-off after three months. The results of our survey should be seen as a stimulus to further sensitize hospital staff to the importance of vaccination and pay attention to anti-S antibody levels monitoring.

## 1. Introduction

In December 2019, the first cases presenting the novel coronavirus (COVID-19) were detected in China, and on 11 March 2020 the World Health Organization (WHO) declared the SARS-CoV-2 pandemic [1]. As of 13 June 2021, outbreaks and sporadic human infections have resulted in more than 175 million confirmed cases and almost 4 million deaths, with a total of over 2 billion vaccine doses administered [2]. SARS-CoV-2 transmission has been high among healthcare workers worldwide, especially in the first wave of the pandemic [3]. Even though vaccination is commonly recognized as the most effective approach, vaccine hesitancy represents a global health issue [4] and a risk of prolonging the pandemic [5]. One of the first vaccines to be approved and used for the vaccination of healthcare workers was Comirnaty (Pfizer/BioNTech, BNT162b2), administered in two doses and highly effective in preventing symptomatic and asymptomatic SARS-CoV-2 infections and COVID-19-related hospitalisations, severe disease, and death [6]. The effectiveness of Comirnaty seems to be proved also for the Delta Variant of Concern (VOC) [7]. To date, no sufficient data have been available to define how long two doses of the vaccine are effective in protecting against infection [8]. For this reason, general vaccination coverage must be supported with studies focused on the measurement of post-vaccination serum antibody concentration. Vaccines currently in distribution induce expression of anti-spike protein antibodies by mRNA technology, such as Pfizer, or the DNA adenovirus vector, such as AstraZeneca [9]. Thus, the presence of anti-S antibodies in serum measures the response to vaccination or to a previous infection [10]. In the absence of exhaustive and clarifying data on the duration of vaccine protection, serum levels of anti-S antibodies should be monitored to understand how long a person will be protected or not [11]. The Istituto Fiorentino di Cura e Assistenza (IFCA) is a private hospital in Florence (Italy). Since the beginning of the pandemic, over 1000 COVID-19 patients were admitted to intensive care and long-term care wards (hospital data, June 2021). In this context, understanding the attitudes of hospital staff workers towards vaccines, identifying reasons for non-adherence, and monitoring anti-S antibody levels at regular intervals could be helpful in promoting effective vaccination strategies and protecting both hospital patients and hospital staff.

## 2. Materials and Methods

The study is a cross sectional investigation. At the time of the vaccination campaign the participants of this study underwent, the Comirnaty (Pfizer/BioNTech, BNT162b2) vaccine was the only one authorized by the European Union for use in hospital employees. In the present study, the period of the vaccination campaign is considered from the beginning of January 2021 to the beginning of March 2021 which involved hospital workers, in addition to people with frailty. Each employee had to book an appointment for the vaccine administration on his own and on a voluntary basis. At the end of March 2021, an anonymous questionnaire was administered to all 368 hospital workers (healthcare workers and non-healthcare workers) (Appendix A). In particular, all the hospital employees, including administrative and technical workers who have not come into contact with patients or infectious materials, were considered as well. The participants joined the study on a voluntary basis after signing informed consent, with no fee for their participation. No personally identifiable information was collected in the questionnaire, such as gender and age. The questionnaire was administered with paper and pen, and it was delivered at the department level by a representative. The principal investigator of the study collected completed questionnaires in each ward. The questionnaire collected information about the type of profession within the hospital (healthcare, administrative or technical worker), data on the previous infection by SARS-CoV-2, whether the workers had been partially, fully or not vaccinated against COVID-19, and if they have been subjected to serological antibody testing for protein S. For people who had not been vaccinated, they were asked the reason for it, by choosing among the following options: unavailability in booking for the vaccine, lack of confidence in the efficacy of the vaccine, presence of contraindications to vaccination, previous contagion, concern about long-term effects. The hospital management decided to assess only healthcare workers’ serological antibody titres against the S-protein for all the people who explicitly accepted to be monitored by signing an informed consent two weeks after the administration of their second dose. Serum sample analyses were performed in a red test tube and BD separator gel was stored at room temperature until arrival in the laboratory where it was centrifuged at 3000 rpm for 15 min and tested. The Roche Elecsys Anti-SARS-CoV-2 S method on Cobas 411 was used for processing the serum samples. The cut-off for defining efficient immunization was 250 IU/mL while those identifying an antibody charge that does not guarantee defense against the virus was <0.8 IU/mL [12]. Processed samples were stored at 2–8 °C for 24 h. The collected information was entered into a database and analyzed using IBM SPSS Statistics (v. 25, Chicago, IL, USA)^TM^. A descriptive analysis was performed on the items collected by means of the questionnaire. For each analysis, an alpha level of 0.05 was considered significant. The items referred to vaccination status (yes/no), previous SARS-CoV-2 infection (yes/no), completion of the second dose (yes/no), having performed blood assay of anti-protein S antibodies (yes/no) were considered as dichotomous variables. The professional role in hospital organization (healthcare, administrative and technical workers), and the causes of non-vaccination, were treated as nominal variables. Frequencies were calculated as absolute numbers and percentages for all variables. The results of the serological analyses were categorized into a dichotomous variable (>250 IU/mL or ≤250 IU/mL). To evaluate the association between profession and vaccination status, profession, and previous COVID-19 infection, vaccinated (yes/no) versus previous SARS-CoV-2 infection (yes/no), Fisher’s exact test was used. The same test was applied to compare below and above (yes/no) the values of the antibody’s titres two weeks and three months after vaccine administration.

## 3. Results

Of 368 questionnaires administered, 345 (93.75%) were filled in and collected. The main results are reported in Table 1.

Most of the respondents were health workers (83.5%), followed by administrative employees (12.8%) and technicians (3.8%). Of the 345 respondents in the study, 297 (86.1%) reported being fully vaccinated (two doses completed). No statistically significant association was observed between vaccination adherence (*p* = 0.98) and professional categories. On the other hand, a statistically significant association (*p* = 0.01) between previous infection and job positions was found: the prevalence of previous infection was higher among healthcare workers (24.3%) and most of them declared to have been infected while working in the hospital (94.6% of all of those who contracted the SARS-CoV-2). Among the unvaccinated subjects (*n* = 48), the majority were healthcare workers (40, 83.3%, *p* = 0.9). In total, 56.2% of the unvaccinated people did not report a previous infection from COVID-19 (*p* = 0.01). Among the reasons for the lack of vaccination, 27.1% (*n* = 13) reported that they did not receive the vaccine due to unavailability of seats during the booking, 29.2% (*n* = 14) declared a previous infection by SARS-CoV-2, 16.7% (*n* = 8) concerned about the long-term effects, 12.5% (*n* = 6) showed distrust against the vaccine, 8.3% (*n* = 4) stated to have received contraindications to the vaccination from a family doctor, and 6.3% (*n* = 3) did not provide any response. Among those who had doubts about the vaccine (*n* = 14), 85.7% were health care workers (*p* = 0.3). Out of 288 vaccinated healthcare workers, 261 reported having undergone serological antibody titre against s-protein. Of these, just 193 (73.9%) gave their consent to use laboratory data for the study and underwent both blood tests. People who had values >250 IU/mL were 94.3% (*n* = 182); those who presented values in the range 0.8–250 IU/mL were 4.7% (*n* = 9) at two weeks. At 3 months, 93.3% (*n* = 180) had values above 250 IU/mL, while 6.2% (*n* = 12) had values in the range 0.8–250 IU/mL. Only one person was found to have a value below the lower cut-off antibody titre, both at two weeks and at three months after receiving their second dose. Moreover, one was below the cut-off of 0.8 IU/mL after two weeks, and then above the same cut-off after three months (*p* = 0.01). In total, 99.5% of vaccinated subjects who underwent the antibody titre (and have given their consent to the processing of data) were still immunized three months after the second dose.

## 4. Discussion

Our results outline a picture of a fairly widespread adherence to COVID-19 vaccination between health and non-health workers, at the beginning of the vaccination campaign. Considering that at that time there was no obligation and that the booking had to be made by the employees on their own, we can hazard a positive outcome from the survey. However, that small proportion of people who are not vaccinated because they are against the vaccine for various reasons arouses trouble and deserves attention. The monitoring of the antibody titre gave us an indication of protection, thanks to the vaccine, from SARS-CoV-2 for at least up to three months. 

The World Health Organization considered vaccine hesitancy a threat to global health [13]. Vaccination adherence is still a debated topic among the general population as well as healthcare professionals. One of the reasons for the mistrust against COVID-19 vaccination lies in the rapid timeline of vaccine development and approval, effectiveness and long-term effects [14]. In the study of Wang et al. [15], low adherence to the vaccine was described among a category of healthcare workers, who raised suspicions on efficacy and side effects. The drivers of this growing problem are complex, context-related, and change in the long run [16]. Kreps et al. [17], for example, reported that greater adherence to the vaccination campaign was recorded in case the vaccine was produced in the same country where the participants resided, approved by national and international scientific bodies. This feature could be useful to better define the communication campaign, both among healthcare professionals and the general population, on the benefits and effects of vaccines, citing all the sources to support the information given and to communicate the updates from those who have authority, both scientific and political. As reported also by Joshi et al. [18], fluctuations related to vaccine adherence and high vaccine hesitation could be an obstacle in reducing the spread of COVID-19.

## 5. Conclusions

Vaccine hesitancy and vaccine refusal represent serious problems in the fight against COVID-19. By identifying the factors that hinder vaccination, it will be possible to plan vaccination campaigns that can lead to overcoming resistance on the part of health and non-health personnel, thus limiting the spread of SARS-CoV-2 and other viruses. Our survey is intended as a further stimulus to promote awareness of hospital staff on the importance of vaccination. Moreover, our study joins that of many other colleagues to monitor anti-S antibody levels over time. Additional results will be reported in a following article as soon as the follow-up of 1 year after the administration of the second dose is reached.

### Limitation of the Study

This study has limitations. First of all, the reported data are localised and may not be representative of other contexts. Secondly, it is a snapshot showing the attitude of the staff shortly after the start of the vaccination campaign. Finally, in order to preserve the privacy of hospital employees, no demographic variables such as gender and age were collected. However, the present study represents a starting point for prospective research of monitoring the level of anti-S antibodies for the duration of immunization given by the Pfizer/BioNTech vaccine.

## Figures and Tables

**Table 1 vaccines-09-01332-t001:** Frequencies in absolute values and percentage of investigated outcomes. *n* = 345.

	Total *n* (%) *	Vaccination Uptake *n* (%) °	Previous SARS-CoV-2 Infection *n* (%) °	Opposition to the Vaccine *n* (%) °
Health workers	288 (83.5%)	248 (86.1%)	70 (24.3%)	12 (4.1%)
Administrative employees	44 (12.7%)	38 (86.4%)	3 (6.8%)	2 (4.5%)
Technical workers	13 (3.8%)	11 (84.6%)	1 (7.7%)	0 (0%)
	345 (100%)	297 (86.1%)	74 (21.4%)	14 (4.1%)

* Column percentage; ° data related to “yes” responses.

## Data Availability

Not applicable.

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
