# Peer review of "Investigating Adherence to COVID-19 Vaccination and Serum Antibody Concentration among Hospital Workers—The Experience of an Italian Private Hospital"

_vaccines, 2021, doi:10.3390/vaccines9111332_

Round 1
Reviewer 1 Report
The manuscript Investigating adherence to COVID-19 vaccination and serum antibody concentration among hospital workers. The experience of an Italian private hospital. is quite interesting and well-written however certain points required corrections by the authors.
Among these points:
1- The authors mentioned in the methodology that the participants were 398 line 64 then they reported out of 368 line 105. which one is correct ?
2- It is highly recommended to add the sample questionnaire in the supplementary
3- Is there any relations related to demographic data ?
4- Is there any reasons given for those who refused to take the vaccine ?
Reviewer 2 Report
The study by Forgeschi et al presents an analysis of vaccine uptake among healthcare, administrative, and technical staff within the context of a single health clinic. The authors show that vaccine hesitancy exists within the healthcare and administrative groups at an approximately similar rate, and although apparently lower within technical workers, this group is far too small to be considered for valid comparison. The authors note that previous COVID infections are expectedly higher among medical staff, however, this does not appear to translate to an overall difference in vaccine uptake. The authors further report high serum antibody titers among the vaccinated group that chose to share such data for publication, however, without pre-vaccination data to compare with or similar data for the unvaccinated population it is difficult to draw any conclusion as to the significance of these numbers. Finally, on page 3, lines 130-131, the authors pose the statement “Only one person was 130 found to have a value below the protective antibody titre, both at two weeks and at three 131 months after receiving second dose.” How was this protective value determined? Correlates of protection for SARS-CoV-2 are not yet established or accepted in the literature, indeed, this property may yet take many months or years to be fully understood. As such I feel this statement is misleading/erroneous and should be removed. Overall, while the authors have made a decent effort to gather data in order to make the case for stronger vaccine encouragement among hospital medical staff, I believe similar, higher quality studies are already widely available in the literature. In light of the weaknesses outlined above I’m unfortunately not certain that this work adds significantly to the scientific record in this area.Author Response
Please see the attachment.

Reviewer 3 Report
In the present work, the authors conducted a study to evaluate vaccination adherence and the anti-S antibodies levels among hospital workers, from January to March 2021. The obtained results could be seen as a stimulus to sensitize hospital staff to the importance of vaccination and pose the attention about anti-S antibody levels monitoring. Due to the importance of these aspects to the field, I suggest that this manuscript must be accepted for publication in Vaccines in the present form.
Round 2
Reviewer 2 Report
I thank the authors for their response comments and efforts to improve the submitted manuscript. Furthermore, I'm in full understanding of their point of view and desire to relay their data gathered within their very specific context of a private medical institution. While I do not doubt efforts put into the work by the authors, or their enthusiasm, unfortunately, my opinion that Vaccines is not the appropriate venue for this work remains unaltered. The Aims & Scope of Vaccines states "Vaccines publishes high quality reviews, regular research papers, communications and case reports." I cannot see that this work, which is observational and not controlled, falls into any of these categories since it is not a review, research, or a case report.
Indeed, that the authors report serological test data, with no control group for comparison, in order to monitor immunity elicited by vaccination highlights a core scientific oversight.
I feel that this manuscript would therefore be better suited to a publication more dedicated to these types of clinical observations and discussion.
